# Weakly Supervised Anomaly Detection via Knowledge-Data Alignment

## ABSTRACT

Anomaly detection (AD) plays a pivotal role in numerous web-based applications, including malware detection, anti-money laundering, device failure detection, and network fault analysis. Most methods, which rely on unsupervised learning, are hard to reach satisfactory detection accuracy due to the lack of labels. Weakly Supervised Anomaly Detection (WSAD) has been introduced with a limited number of labeled anomaly samples to enhance model performance. Nevertheless, it is still challenging for models, trained on an inadequate amount of labeled data, to generalize to unseen anomalies. In this paper, we introduce a novel framework Knowledge-Data Alignment (KDAlign) to integrate rule knowledge, typically summarized by human experts, to supplement the limited labeled data. Specifically, we transpose these rules into the knowledge space and subsequently recast the incorporation of knowledge as the alignment of knowledge and data. To facilitate this alignment, we employ the Optimal Transport (OT) technique. We then incorporate the OT distance as an additional loss term to the original objective function of WSAD methodologies. Comprehensive experimental results on five real-world datasets demonstrate that our proposed KDAlign framework markedly surpasses its state-of-the-art counterparts, achieving superior performance across various anomaly types.

## KEYWORDS

Anomaly Detection; Knowledge-Data Alignment; Weakly Supervised Learning

### ACM Reference Format:

Anonymous Author(s). 2018. Weakly Supervised Anomaly Detection via Knowledge-Data Alignment. In *Proceedings of Make sure to enter the correct conference title from your rights confirmation emai (Conference acronym 'XX).* ACM, New York, NY, USA, 12 pages. https://doi.org/XXXXXXX.XXXXXXX

## 1 INTRODUCTION

Anomaly detection (AD), aiming at identifying patterns or instances that deviate significantly from the expected behavior or normal patterns, is crucial to extensive web-based applications including malware detection [29], anti-money laundering [33], device failure detection [50], network fault analysis [64]. Given that labeled anomaly data is typically scarce or costly to acquire, unsupervised

A note.

Permission to make digital or hard copies of all or part of this work for personal or classroom use is granted without fee provided that copies are not made or distributed for profit or commercial advantage and that copies bear this notice and the full citation on the first page. Copyrights for components of this work owned by others than ACM must be honored. Abstracting with credit is permitted. To copy otherwise, or republish, to post on servers or to redistribute to lists, requires prior specific permission and/or a fee. Request permissions from permissions@acm.org.

*Conference acronym 'XX, June 03–05, 2018, Woodstock, NY*

© 2018 Association for Computing Machinery.
ACM ISBN 978-1-4503-XXXX-X/18/06...$15.00
https://doi.org/XXXXXXX.XXXXXXX

methodologies that operate on entirely unlabeled data have gained widespread use. However, in the absence of supervision, these models may incorrectly classify noisy or unrelated data as anomalies, leading to high detection errors.

To alleviate the above issue, Weakly Supervised Anomaly Detection (WSAD) has been proposed to enhance detection accuracy with limited labeled anomaly samples and a large amount of unlabeled data [27], shown in Fig. 1(a). Early studies use unsupervised AD algorithms as feature extractors and learn a supervised classifier with label data [25, 35, 40, 49, 55]. With the development of deep learning, most recent studies focus on end-to-end frameworks that build on multilayer perceptron, autoencoder, and generative adversarial networks, to directly map input data to anomaly scores [1, 42, 43, 65]. Nevertheless, models trained on insufficient labeled data fail to generalize to novel anomalies or anomalies not observed during training time [26, 27]. Although several works have employed active learning or reinforcement learning to reduce the cost of obtaining anomaly labels, they still require an initial set of labeled data to start the learning process, which can be costly and time-consuming [44, 62].

In this work, we propose to incorporate rule knowledge, which is often derived or summarized by human experts [6, 34, 37, 60, 64], similar to label annotation but has been largely overlooked, to help complement the limited labeled data, as shown in Fig. 1(b). Although rules are high-quality and accessible in practice [34, 37], incorporating them is non-trivial for three reasons: (1) knowledge representation: rules are generally represented by if/else statements [34, 37]. In the representation space, rules and data lack a direct correlation [58, 64], making them unsuitable for directly training the WSAD models; (2) knowledge-data alignment: intuitively, if two rules are close then their corresponding data samples should be also close [6]. For example, in anti-money laundering, a group of fraudsters may possess similar patterns and thus have similar data representations [6, 34, 37]. Usually, these fraudsters will be detected by identical or similar if/else rule statements in anti-money laundering systems [6]. In this work, we reformulate the knowledge incorporation process as the knowledge-data alignment and supplement the traditional data-only optimizations; (3) noisy knowledge: typically, rules are not always accurate [6, 34], thus directly aligning them with data may involve noises and resulting in a performance drop. It is still challenging to ensure the model's performance under noisy rules.

To address the above issues, we propose a novel framework for Weakly Supervised Anomaly Detection via **K**nowledge-**D**ata **Align**ment (**KDAlign**). KDAlign expects to align knowledge and data to complement the data distribution. For the first challenge, KDAlign employs a knowledge encoder to map the rules into an embedding space, thereby allowing knowledge to correlate with data in the numerical domain. For the second and third challenges, KDAlign leverages the Optimal Transport (OT) technique to align

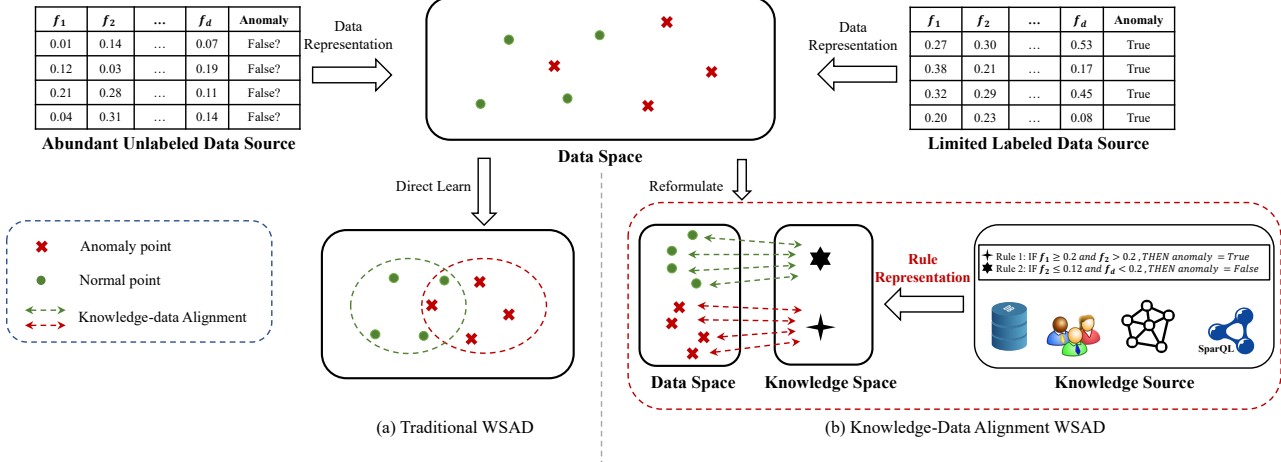

**Figure 1: Comparison between traditional WSAD approach (a) and our proposed knowledge-data alignment WSAD framework (KDAlign) (b). We can find that the traditional WSAD mainly focuses on learning from limited labeled data, while our proposed framework introduces knowledge as extra information to supplement limited labeled via knowledge-data alignment. Note that the samples in the unlabeled data source are usually regarded as normal samples, though the unlabeled data may be contaminated by noise [26, 41].**

knowledge and data. The primary strength of OT lies in its inherent flexibility. It autonomously determines the optimal transport pairings between knowledge and data, thus establishing an intrinsic connection [39, 54]. Specifically, the OT method intrinsically forms a robust framework, enabling a geometrically faithful comparison of probability distributions and facilitating the information transfer between distinct distributions [21]. Regarding noisy knowledge, when a sample matches a noisy rule, the distance of that sample to some other closely related rules will be farther, resulting in an increased OT distance penalty. To ensure global optimality, the OT distance between this sample and the noisy rule will be constrained by other correct rules, thereby ensuring the performance of KDAlign.

To sum up, our contributions are three-fold:

- To the best of our knowledge, this is the first work to incorporate rule knowledge into WSAD, effectively complementing the limited labeled data.
- We propose a novel Knowledge-Data Alignment Weakly Supervised anomaly detection framework (KDAlign).
- The experimental results on five public WSAD datasets indicate our proposed KDAlign are superior to all the competitors. Furthermore, KDAlign achieves strong performance improvements even with 20% noisy rule knowledge.

## 2 PRELIMINARY

### 2.1 Rule Knowledge and Logical Formulae

In this paper, we focus on rule knowledge (if/else). This choice stems from the high-quality and accessible in practice of rules — they present explicit conditions and outcomes. Such transparency allows individuals to understand anomaly and normal data easily. To avoid the potential overlaps among different rules, we adopt a precise

knowledge statement format named Logical Formulae. Concretely, logical statements provide a flexible declarative language for expressing structured knowledge (e.g., rule knowledge). In this paper, we focus on **propositional logic**, where a **proposition** $p$ is a statement which is either **True** or **False** [32]. A statement (proposition) consists of a subject, predicate and object. It can also be regarded as a ground clause that does not contain any variables [16]. A *propositional formula* $f$ is a compound of propositions connected by logical connectives [7, 58], e.g., $\neg, \wedge, \vee, \Rightarrow$. Also, a propositional formula is equal to a grounding first-order logic formula. In the subsequent content, we use $\mathbf{F} = \{f_1, \ldots, f_s\}$ to represent a set of propositional formulae, where $f_i$ is a propositional formula and $s$ is the number of propositional formulae. The concrete proposition formats designed for rule knowledge of AD are introduced in Section 3.

### 2.2 Problem Statement

Given a training dataset $\mathbf{X} = \{\mathbf{x}_1, \mathbf{x}_2, \cdots, \mathbf{x}_n, \mathbf{x}_{n+1}, \cdots, \mathbf{x}_{n+m}\}$, with $\mathbf{x}_i \in \mathbb{R}^d$, where $\mathbf{X}_U = \{\mathbf{x}_1, \mathbf{x}_2, \cdots, \mathbf{x}_n\}$ is a large unlabeled dataset and $\mathbf{X}_A = \{\mathbf{x}_{n+1}, \mathbf{x}_{n+2}, \cdots, \mathbf{x}_{n+m}\}$ $(n \ll m)$ is a small set of labeled anomaly examples that often can not cover every possible class of anomaly, a WSAD model $\mathcal{M}$ is first trained on $\mathbf{X}$ to output anomaly score $\mathbf{O} := \mathcal{M}(\mathbf{X}) \in \mathbb{R}^{m \times 1}$, where higher scores indicate a higher likelihood of an abnormal sample. The unlabeled dataset $\mathbf{X}_U$ is usually assumed as normal data, though it may be contaminated by some anomalies in practice [41, 42]. Thus, the trained $\mathcal{M}$ is required to be robust *w.r.t.* such anomaly contamination. Based on the trained model, we need to predict on the unlabeled test dataset $\mathbf{X}_T \in \mathbb{R}^{q \times d}$, so to return $\mathbf{O}_T := \mathcal{M}(\mathbf{X}_T) \in \mathbb{R}^{q \times 1}$. In our work, we introduce a set of rule knowledge represented by propositional formulae $\mathbf{F}$, as extra information, to supplement data and then training a WSAD models on $\{\mathbf{X}, \mathbf{F}\}$.

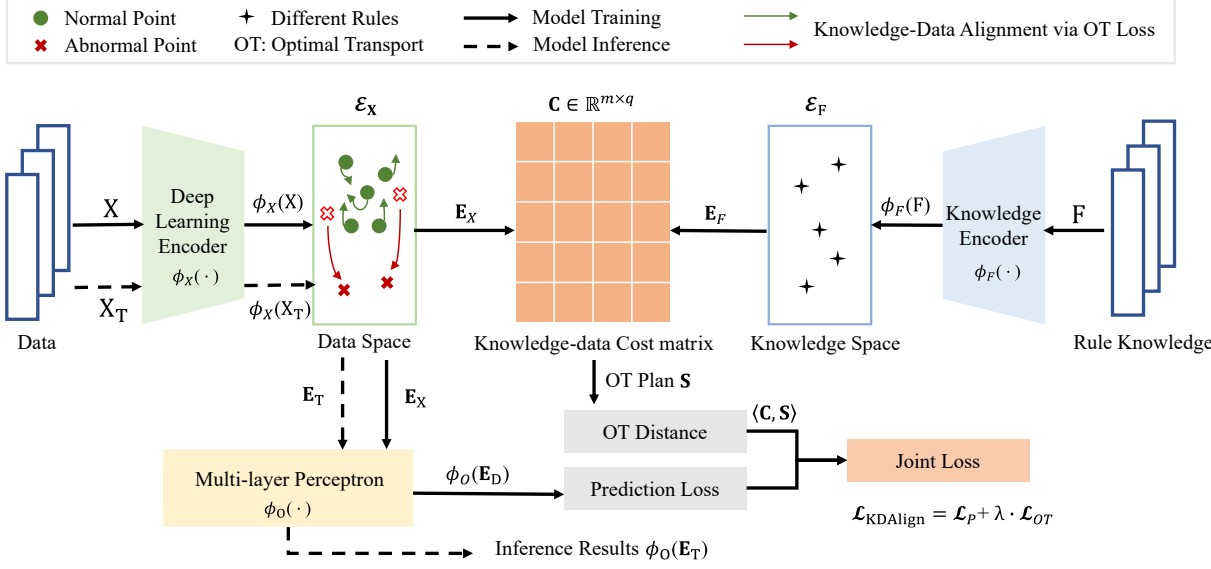

**Figure 2: Knowledge-data alignment WSAD framework. During the training phase, we firstly use $\phi_X$ and $\phi_F$ to map X and F to two separate embedding spaces and then leverage Optimal Transport (OT) techniques to compute the cost matrix C, thereby obtaining the OT plan S. Next, we compute OT distance $\langle C, S \rangle$ and add it as a loss term to the prediction loss term, forming a joint loss. Finally, we utilize the joint loss to train $\phi_X(\cdot)$ and $\phi_O(\cdot)$, aligning knowledge and data for incorporating knowledge. In the inference phase of the model, the test data directly yields results by $\phi_X$ and $\phi_O$. In the data space, both the abnormal and normal points can be aligned via OT.**

## 3 METHODOLOGY

### 3.1 Overview

Figure 2 provides an overview of our proposed **K**nowledge-**D**ata **Align**ment WSAD framework (**KDAlign**). First, we utilize a deep learning encoder and a knowledge encoder to project data **X** and knowledge **F** into data embedding space $\mathcal{E}_X$ and knowledge embedding space $\mathcal{E}_F$, respectively, making operations between knowledge and data possible. The deep learning encoder could be based on a multi-layer perceptron, autoencoder, or ResNet-like architecture. The knowledge encoder is a multi-layer graph convolutional network. Second, within the embedding space, we align data and knowledge via the Optimal Transport Technique (OT) and then leverage the alignment result to derive an OT loss term. The OT loss term is subsequently used for introducing the knowledge into deep learning models. Besides, we give an analysis of why KDAlign has the potential to alleviate the noisy knowledge issue. Third, we jointly leverage the OT loss and the prediction loss to train a deep-learning model, expecting to learn better data representations for **X** and improve the model performance, where the original loss is computed by the output of the multi-layer perceptron and labels. Then, the trained model can be used for inferring unlabeled test data.

### 3.2 Representation Framework

#### 3.2.1 *Data Representation*. Given the training dataset **X** with $m$ samples, we use a deep learning encoder to project data into a high-dimensional embedding space, generating the corresponding

data embedding set $\mathbf{E}_X = \{e_1, \ldots, e_m\}$, where $m$ is the number of samples. The process is shown by Equation 1

$$\mathbf{E}_X = \phi_X(\mathbf{X}) \in \mathbb{R}^{m \times h}, \tag{1}$$

where $\phi_X$ is a deep learning encoder, and $h$ is the dimension of the Data Space $\mathcal{E}_X$ defined by $\mathbf{E}_X$.

#### 3.2.2 *Knowledge Representation*. In order to enable knowledge and data to be operated, we also consider representing knowledge in a high-dimensional embedding space, but embedding knowledge successfully entails rendering it into a format amenable for processing by deep learning methods. Given $s$ if/else statements, we first contemplate transforming their formats into propositional logic and then generate a knowledge set **F** with $s$ propositional formulae, which is drawn inspiration from knowledge embedding of logical formulae [57].

Next, we provide an example to illustrate how to transform the if/else statement into propositional logic: Given an if/else statement '$If\ attr_1 > 5\ and\ attr_2 = 0$, $then\ anomaly\ is\ True$', where $attr_1$ and $attr_2$ correspond to two attributes of the sample, three propositions $\{p_1 = (attr_1 > 5), p_2 = (attr_2 = 0), p_3 = (anomaly\ is\ True)\}$ and a proposition formula $f = \{p_1 \wedge p_2 \Rightarrow p_3\}$ can be generated. The subject, object, and predicate constituting the three propositions are, respectively, $subject = \{attr_1, attr_2, anomaly\}$, $object = \{5, 0, True\}$, $predicate = \{>, =, is\}$.

Building upon the example above, we provide definitions for the subject, object, and predicate in propositional formulae used to describe if/else statements: The subject comprises attribute names

present in the sample and the word 'anomaly'; the object encompasses attribute thresholds, words 'True', and 'False'; the predicate consists of numeric relational symbols and the word 'is'.

Based on the transformation strategy, we convert if/else statements into a set of propositional formulae $\mathbf{F} = \{f_1, ..., f_s\}$. Subsequently, each propositional formula in $\mathbf{F}$ is transformed into a graph structure, and then a multi-layer graph convolutional network [57] is constructed as a knowledge encoder to project propositional formulae into a high-dimensional embedding space named Knowledge Space and generate the knowledge embedding set $\mathbf{E}_F = \{e_1, ..., e_s\}$, where $s$ is the number of propositional formulae, shown in Equation 2

$$\mathbf{E}_F = \phi_F(\mathbf{F}) \in \mathbb{R}^{s \times h}, \tag{2}$$

where $\phi_F(\cdot)$ is a multi-layer graph convolutional network, and $h$ is the dimension of the Knowledge Space $\mathcal{E}_F$ defined by $\mathbf{E}_F$. Since the dimensions of $\mathcal{E}_F$ and $\mathbf{E}_X$ are both $h$, knowledge and data lie in the same dimensional space. Additionally, $\phi_F(\cdot)$ is trained before the training of the deep learning model $\phi_X$. The details of the knowledge encoder are shown in the appendix.

## 3.3 Knowledge-Data Alignment

### 3.3.1 Definition.
Given a knowledge set $\mathbf{F}$ and a limited labeled training dataset $\mathbf{X}$, without any observed knowledge-data correspondences, the alignment algorithm returns aligned knowledge-data pairs $M = \{(f_i, x_j) | (f_i, x_j) \in F \times X\}$.

### 3.3.2 Optimal Transport.
To resolve knowledge-data alignment, we leverage the OT technique that has been widely applied in various domains for alignment, such as in image and graph domains [3, 54, 56, 63]. We follow the Kantorovich formulation [20, 61], which can be formally defined in terms of two distributions and a cost matrix as follows

**Definition 3** (Optimal Transport). Given two sets of observations $O_1 = \{o_1, ..., o_{n_1}\}, O_2 = \{o_1, ..., o_{n_2}\}$, there are two discrete distributions $\mu, v$ defined on probability simplex $\Delta_1, \Delta_2$, where $n_1$ is the number of $O_1$, and $n_2$ is the number of $O_2$. Then, a cost matrix $\mathbf{C} \in \mathbb{R}^{n_1 \times n_2}$ is computed for measuring the distance between all pairs $(f_i, x_j) \in \Delta_1 \times \Delta_2$ across two distributions. The OT problem aims to find an OT plan $\mathbf{S} \in \Pi(\mu, v)$ between $\mu$ and $v$ that minimizes the expected cost over the coupling as follows:

$$\min_{\mathbf{S} \in \Pi(\mu,v)} \sum_{f_i, x_j} \mathbf{C}(f_i, x_j) \mathbf{S}(f_i, x_j) = \min_{\mathbf{S} \in \Pi(\mu,v)} \langle \mathbf{C}, \mathbf{S} \rangle,$$

$$\text{s.t.} \quad \mathbf{S}(f_i, x_j) \geq 0, \ for \ all \ i \ and \ j, \tag{3}$$

$$\sum_{i=1}^{n_1} \mathbf{S}(f_i, x_j) = \mu_i, \sum_{j=1}^{n_2} \mathbf{S}(f_i, x_j) = v_j,$$

where $\mathbf{S}$ is the OT plan, $\Pi(\mu, v)$ is the probabilistic coupling between $\mu$ and $v$ (i.e., all the available transport plan between $\mu$ and $v$), $\langle \cdot, \cdot \rangle$ is inner product, and corresponding $\langle \mathbf{C}, \mathbf{S} \rangle$ is the Wasserstein distance between $\mu$ and $v$. In this paper, to efficiently solve the OT problem, we employ the Sinkhorn-Knopp algorithm [10, 20]. The objective of the Sinkhorn-Knopp algorithm is to approximate

the computation of the Wasserstein distance, enabling efficient computation of Wasserstein distance, particularly in high-dimensional or large-scale scenarios.

In knowledge-data alignment, we regard $\mu$ as the knowledge distribution defined by $\mathbf{F}$ and $v$ as the data distribution defined by $\mathbf{X}$, and then $\mathbf{S}(f_i, x_j)$ indicates the matching score between $f_i$ in the knowledge set $F$ and $x_j$ in $X$. The alignment $M$ can be derived from $\mathbf{S}$:

$$M = \arg\max_{M \in \mathbb{M}} \sum_{(f_i, x_j) \in M} \mathbf{S}(f_i, x_j), \tag{4}$$

where $\mathbb{M}$ is the set of all legit alignments, $i = 1, ..., s$, and $j = 1, ..., m$.

### 3.3.3 OT Loss For Weakly Supervised Anomaly Detection.
Obtaining the knowledge-alignment results, we further describe how to utilize the alignment results to benefit WSAD. Firstly, we compute the knowledge-data cost matrix $\mathbf{C} \in \mathbb{R}^{s \times m}$ for measuring the distance between all pairs $(\phi_F(f_i), \phi_X(x_j)) \in \mathcal{E}_F \times \mathcal{E}_X$ across two distribution spaces, which describes the discrepancy between the data embeddings and the knowledge embeddings. Secondly, we can compute the OT plan $S$ and the knowledge-data alignment $M$ by Equations 3 and 4, respectively. Thirdly, we also compute the OT distance, which quantifies the minimum cost required to transform one probability distribution into another, by $\langle \mathbf{C}, \mathbf{S} \rangle$. Finally, the OT distance is used for training the deep learning model.

**Analysis On Noisy Knowledge Alleviation.** Firstly, we need to clarify the effects of noisy rules on WSAD. Since rules and data are matched one-to-one, when noisy rules emerge, it directly leads to detection errors. We would naturally assume that introducing noisy rule information into data embeddings can also impact the performance of the WSAD model. However, benefiting from the OT technique, which takes a global perspective to align rules and data, our proposed KDAlign framework would not be obviously influenced by noisy rules. This is because introducing too much noisy rule information can lead to excessive transport distances between the data and other relevant rules, resulting in a suboptimal transport plan. Therefore, to provide the optimal transport plan, the incorporation of noisy rule knowledge is constrained by other correct rules, thereby alleviating the impact of noisy knowledge.

## 3.4 Model Training and Inference

**Model Training.** In addition to deep learning encoders for embedding data, we also employ a multi-layer perceptron (MLP) $\phi_O(\cdot)$ to output anomaly scores or classification results based on data embeddings. During the training process, to effectively leverage alignment results, we introduce the OT distance between knowledge and data as a loss term $\mathcal{L}_{OT}$ added to the prediction loss function $\mathcal{L}_P$ computed by output and sample labels, rather than as a regularization term. This is mainly because the OT distance calculated by the Sinkhorn-Knopp algorithm is differentiable. Concretely, this addition introduces an auxiliary objective that allows both the deep learning encoder and MLP to simultaneously update parameters based on $\mathcal{L}_P$ and $\mathcal{L}_{OT}$, effectively incorporating knowledge information into data embeddings. The joint loss function $\mathcal{L}_{KDAlign}$ is shown by Equation 5

$$\mathcal{L}_{KDAlign} = \mathcal{L}_P + \lambda \cdot \mathcal{L}_{OT}, \tag{5}$$

where $\mathcal{L}_P$ is the prediction loss, and $\mathcal{L}_{OT}$ is computed by $\langle C, S \rangle$, and $\lambda$ is the trade-off factor of $\mathcal{L}_{OT}$. From another perspective, the OT loss offers a targeted optimization direction, thereby effectively incorporating knowledge information and enhancing the model performance. In addition, the prediction loss could also be alternated by other losses for AD (e.g., deviation loss [43] and the specially designed deviation loss [65].

***Model Inference.*** The trained deep learning encoder $\phi_X$ and MLP $\phi_O$ comprise the WSAD model $\mathcal{M}$, which is used for inference on test dataset by Equation 6

$$\mathcal{M}(X_T) = \phi_O(\phi_X(X_T)). \tag{6}$$

where $\phi_X$ is the trained deep learning encoder, $\phi_O$ is the trained MLP, $\mathcal{M}$ is the trained WSAD model, and $X_T$ is the test dataset.

## 4 EXPERIMENTS

In this section, we study the experimental results of our proposed method and baselines to answer three research questions:

- **RQ1.** How effective is the proposed KDAlign framework that incorporates knowledge compared with representative baselines in WSAD?
- **RQ2.** How important is the Knowledge-data Alignment in KDAlign?
- **RQ3.** How does noisy knowledge impact the KDAlign?

## 4.1 Experimental Setup

***Datasets.*** We conduct experiments on five real-world datasets [23, 26, 53]. The **YelpChi** dataset[47] is used for finding anomalous reviews which unjustly promote or demote certain products or businesses on Yelp.com. The **Amazon** dataset[38] seeks to identify the anomalous users paid to write fake product reviews under the Musical Instrument category on Amazon.com. The **Cardiotocography** dataset[2] targets to detect the pathologic fetuses according to fetal cardiotocographies. The **Satellite** dataset [52] is collected for distinguishing anomalous satellite images according to multi-spectral values of pixels in 3x3 neighbourhoods. The **SpamBase** dataset[24] is leveraged to decide spam e-mails on e-mail systems. The descriptions of the five datasets are shown in Table 1.

***Metrics.*** We choose two widely used metrics to evaluate the performance of all the methods[17, 23, 53], namely **AUPRC (Area Under the Precision-Recall Curve)**, and **Rec@K (Recall at k)**. AUPRC is the area beneath the Precision-Recall curve at different thresholds. **AUPRC** can be calculated by the weighted mean of precisions at each threshold, where the increase in recall from the previous threshold serves as the weight. **Rec@K** is determined by calculating the recall of the true anomalies among the top-k predictions that the model ranks with the highest confidence. We set the value of k as the number of actual outliers in the test dataset. It is noteworthy that in this specific scenario, Rec @K is equivalent to both precision at k and the F1 score at k (**F1@K**).

**Table 1: Data description of five datasets used in our experiments. Rule-Detect denotes the number of samples that match rules. Rate is computed by #Rule-Detect/#Label.**

| Name | Size | #Feature | #Rule | #Label | #Rule-Detect | Rate(%) |
|---|---|---|---|---|---|---|
| Amazon | 11944 | 25 | 20 | 821 | 431 | 52.0 |
| Carditocography | 2114 | 21 | 14 | 466 | 281 | 60.0 |
| Satellite | 6435 | 36 | 23 | 2036 | 1015 | 50.0 |
| SpamBase | 4207 | 57 | 21 | 1679 | 968 | 58.0 |
| YelpChi | 45954 | 32 | 88 | 6678 | 1475 | 22.0 |

***Baselines.*** We compare the proposed method with the following baselines and give brief descriptions. The first three are typical AD methods. The rest of them are representative of WSAD methods.

- **k-Nearest Neighbors (KNN)** [9]. A classification method based on the k nearest neighbors in the training set.
- **Support Vector Machine (SVM)** [8]. A classification method based on maximum margin.
- **Decision Tree (DT)** [5]. A classification method based on tree structure, and every decision path define an if/else statement.
- **DeepSAD** [49]. A deep semi-supervised one-class method that enhances the unsupervised DeepSVDD.
- **REPEN** [40]. A neural network based model that utilized transformed low-dimensional representation for random distance based detectors.
- **DevNet** [43]. A neural network based model trained by deviation loss.
- **PReNet** [42]. A neural network based model that defines a two-stream ordinal regression to learn the relation of instance pairs.
- **FeaWAD** [65]. A neural network based model that incorporates the network architecture of DAGMM [66] with the deviation loss of DevNet.
- **ResNet** [22]. ResNet-like architecture turns out to be a strong baseline [26].

***Parameter And Implementation Details.*** Firstly, acquiring knowledge is essential. It is worth noticing that the five datasets do not provide rules, and due to industrial security and privacy issues, obtaining well-defined rules directly is challenging. Therefore, we need to simulate the rules of these datasets to acquire knowledge. In our experiments, we train several decision tree models for each dataset using additional labels, and then extract the decision paths from the decision tree models as our if/else rules. In WSAD, anomaly samples are unbalanced and important, so we focus on the decision paths used for anomaly samples. A more detailed description of knowledge acquisition can be referred to the appendix. Secondly, we divide each dataset into a training set, a validation set, and a test set according to the scale of 7:1:2. To ensure that the rules really provide extra information (e.g., unseen anomaly scenarios) to supplement limited labeled samples, we delete the anomaly samples that match rules from the training set. Besides, for each training set, we only retain 10 labeled anomaly samples, treating the rest of the anomalies and all normal samples as unlabeled data, with the default label being normal samples. Besides, we also consider another three training settings with 1, 3, and 5 labeled anomaly samples.

**Table 2: Performance comparison between representative baselines and KDAlign w.r.t. AUPRC and F1@K. The best results are in bold.**

| Model | Amazon | | Cardio | | Satellite | | SpamBase | | YelpChi | |
|---|---|---|---|---|---|---|---|---|---|---|
| | PRC | F1@K | PRC | F1@K | PRC | F1@K | PRC | F1@K | PRC | F1@K |
| KNN | 0.074 | 0.071 | 0.370 | 0.333 | 0.326 | 0.359 | 0.419 | 0.406 | 0.145 | 0.151 |
| SVM | 0.127 | 0.013 | 0.654 | 0.570 | 0.312 | 0.296 | 0.357 | 0.283 | 0.130 | 0.114 |
| DT | 0.078 | 0.065 | 0.261 | 0.226 | 0.320 | 0.357 | 0.455 | 0.431 | 0.145 | 0.149 |
| DeepSAD | 0.137 | 0.206 | 0.253 | 0.312 | 0.604 | 0.509 | **0.762** | **0.686** | 0.187 | 0.215 |
| KDAlign-DeepSAD | **0.201** | **0.252** | **0.420** | **0.462** | **0.617** | **0.535** | 0.731 | 0.637 | **0.207** | **0.248** |
| REPEN | 0.116 | 0.039 | 0.452 | 0.473 | **0.726** | 0.648 | 0.605 | 0.589 | **0.245** | **0.283** |
| KDAlign-REPEN | **0.289** | **0.290** | **0.631** | **0.559** | 0.720 | **0.658** | **0.608** | **0.606** | 0.181 | 0.204 |
| DevNet | 0.250 | 0.316 | 0.266 | 0.258 | 0.647 | **0.543** | 0.416 | 0.420 | 0.186 | 0.211 |
| KDAlign-DevNet | **0.487** | **0.626** | **0.490** | **0.516** | **0.676** | 0.533 | **0.517** | **0.526** | **0.195** | **0.218** |
| PReNet | 0.580 | 0.574 | 0.602 | 0.591 | 0.331 | 0.303 | 0.848 | 0.783 | 0.174 | 0.200 |
| KDAlign-PReNet | **0.728** | **0.716** | **0.671** | **0.624** | **0.677** | **0.604** | **0.850** | **0.783** | **0.181** | **0.205** |
| FeaWAD | 0.779 | 0.768 | 0.622 | 0.591 | 0.322 | 0.418 | 0.749 | 0.620 | 0.184 | 0.220 |
| KDAlign-FeaWAD | **0.789** | **0.794** | **0.664** | **0.624** | **0.601** | **0.555** | **0.776** | **0.734** | **0.216** | **0.247** |
| ResNet | 0.770 | 0.729 | 0.566 | 0.612 | 0.352 | 0.384 | 0.756 | 0.706 | 0.183 | 0.203 |
| KDAlign-ResNet | **0.848** | **0.768** | **0.659** | **0.656** | **0.604** | **0.594** | **0.770** | **0.734** | **0.209** | **0.262** |

Our implementation of SVM, KNN, and DT is consistent with the APIs of Sklearn [45]. We keep the default settings of SVM, KNN, and DT given by Sklearn. For representative WSAD methods, DeepSAD, REPEN, DevNet, PReNet, and FeaWAD are consistent with the Benchmark DeepOD [59], and ResNet is implemented based on the design for AD from [22]. The default optimizer of each baseline is Adam [30]. To apply our proposed KDAlign framework, we make slight adjustments to the representative WSAD methods. Concretely, during the forward propagation of these methods, in addition to returning the output of the final layer, they also return the sample hidden representations from the layer before the last one. All the models are tuned to the best performance on the validation set. Our codes are released at https://github.com/KDAlignForWWW2024/KDAlign.

## 4.2 Performance Comparison (RQ1)

Table 2 shows the model performance on 5 datasets w.r.t AUCPR and F1@K. Each dataset contains 10 labeled anomalies. Above all, we verify the effectiveness of the KDAlign framework on various deep learning-based WSAD baselines. The KDAlign based AD methods we proposed generally outperform the corresponding baselines.

Specifically, we have the following observations:

- We find that typical anomaly detection methods, including KNN, SVM, and DT, struggle when the number of labeled anomalies is extremely sparse. The SVM method shows good results on the Cardiotocography dataset, which might be coincidental.
- We observe that while representative WSAD methods demonstrate impressive performance on certain datasets, they invariably have weak performance on one or more datasets. For instance, DeepSAD outperforms most baselines on the SpamBase dataset but underperforms on the Amazon dataset. Based on the

characteristics of these five datasets, the discrepancy may be due to the higher feature count in SpamBase and the lower feature count in the Amazon dataset. This is because DeepSAD focuses on anomaly feature representation learning [27]. REPEN method outperforms all other methods on the YelpChi dataset, but falls short on both the Amazon and Cardiotocography datasets. We surmise that this is because REPEN is an unsupervised anomaly feature representation learning method [27], and datasets like Amazon and Cardiotocography neither offer as many samples as YelpChi nor as many features as Satellite and SpamBase. The performance of the DevNet method on the Amazon, Cardiotocography, and SpamBase datasets is not satisfactory. This is mainly because the labeled anomalies available for these three datasets cover a limited range of anomaly scenarios. As the DevNet approach focuses on Anomaly Score Learning [27], the scores it learns fail to distinguish between normal and anomalous samples.

- We find that methods like PReNet and FeaWAD, both belonging to the anomaly score learning [27], generally outperform DevNet and other baseline methods across all datasets. We attribute this promising performance primarily to the design of the PReNet and FeaWAD. PReNet method takes anomaly-anomaly, anomaly-unlabeled, and unlabeled-unlabeled instance pairs as input, and learns pairwise anomaly scores by discriminating these three types of linear pairwise interactions. This is an augmentation process of data distribution for existing labeled anomalies, which subsequently aids in the learning of the final anomaly scores. The autoencoder architecture of FeaWAD is capable of mapping limited labeled samples to a latent space, thereby extending the distribution of these sparsely labeled anomalies and improving the learned Score distributions.

**Table 3: Performance comparison between representative baselines and KDAlign under the setting of 1, 3, or 5 labeled anomalies w.r.t AUPRC. '-' indicates that the PReNet model can not handle setting of only 1 labeled anomaly samples.**

| Model | Amazon | | | Cardio | | | Satellite | | | SpamBase | | | YelpChi | | |
|---|---|---|---|---|---|---|---|---|---|---|---|---|---|---|---|
| | 1 | 3 | 5 | 1 | 3 | 5 | 1 | 3 | 5 | 1 | 3 | 5 | 1 | 3 | 5 |
| KNN | 0.065 | 0.065 | 0.080 | 0.236 | 0.253 | 0.286 | 0.323 | 0.319 | 0.319 | 0.416 | 0.416 | 0.414 | 0.145 | 0.145 | 0.145 |
| SVM | 0.238 | 0.124 | 0.093 | 0.290 | 0.304 | 0.293 | 0.349 | 0.305 | 0.313 | 0.437 | 0.425 | 0.536 | 0.148 | 0.150 | 0.165 |
| DT | 0.065 | 0.065 | 0.113 | 0.229 | 0.244 | 0.286 | 0.318 | 0.318 | 0.318 | 0.426 | 0.429 | 0.433 | 0.145 | 0.145 | 0.145 |
| DevNet | 0.071 | 0.103 | 0.100 | 0.267 | 0.269 | 0.286 | **0.722** | **0.717** | **0.725** | 0.416 | 0.416 | 0.416 | **0.167** | 0.169 | 0.169 |
| KDAlign-DevNet | 0.071 | **0.747** | **0.581** | **0.528** | **0.436** | **0.502** | 0.449 | 0.641 | 0.686 | **0.489** | **0.461** | **0.494** | 0.156 | **0.200** | **0.195** |
| PReNet | - | 0.152 | 0.488 | - | 0.390 | 0.626 | - | 0.344 | 0.266 | - | 0.689 | **0.819** | - | 0.164 | 0.161 |
| KDAlign-PReNet | - | **0.617** | **0.623** | - | **0.656** | **0.558** | - | **0.748** | **0.767** | - | 0.741 | 0.817 | - | **0.176** | **0.207** |
| DeepSAD | 0.129 | 0.070 | 0.104 | 0.355 | 0.248 | 0.259 | **0.732** | **0.739** | 0.670 | **0.710** | 0.668 | 0.602 | 0.144 | 0.199 | 0.197 |
| KDAlign-DeepSAD | **0.347** | **0.514** | **0.271** | **0.639** | **0.480** | **0.517** | 0.717 | 0.480 | **0.735** | 0.699 | **0.778** | **0.779** | 0.144 | **0.208** | **0.215** |
| REPEN | 0.116 | 0.116 | 0.116 | 0.452 | 0.452 | 0.452 | **0.726** | 0.703 | 0.703 | 0.605 | 0.605 | 0.605 | **0.245** | **0.244** | **0.245** |
| KDAlign-REPEN | **0.289** | **0.289** | **0.289** | **0.631** | **0.631** | **0.631** | 0.720 | **0.720** | **0.720** | **0.608** | **0.608** | **0.608** | 0.181 | 0.180 | 0.209 |
| FeaWAD | 0.692 | 0.274 | 0.734 | 0.545 | **0.527** | **0.622** | 0.582 | 0.570 | 0.686 | **0.697** | 0.717 | 0.756 | 0.170 | 0.177 | 0.212 |
| KDAlign-FeaWAD | **0.738** | **0.611** | **0.778** | **0.654** | 0.489 | 0.551 | **0.751** | **0.769** | 0.529 | 0.615 | **0.765** | **0.775** | **0.212** | 0.193 | **0.240** |
| ResNet | 0.785 | 0.629 | 0.777 | 0.590 | **0.663** | **0.700** | 0.328 | 0.359 | 0.353 | **0.625** | 0.601 | 0.637 | 0.183 | 0.175 | 0.175 |
| KDAlign-ResNet | **0.805** | **0.744** | **0.834** | **0.642** | 0.613 | 0.691 | **0.696** | **0.676** | **0.699** | 0.601 | **0.621** | **0.668** | **0.201** | **0.207** | **0.207** |

**Table 4: AUPRC and F1@K results of Ablation Study. KD- represents the WSAD method incorporated knowledge without knowledge-data alignment.**

| Labeled Anomalies | Model | Amazon | | Cardio | | Satellite | | SpamBase | | YelpChi | |
|---|---|---|---|---|---|---|---|---|---|---|---|
| | | PRC | F1@K | PRC | F1@K | PRC | F1@K | PRC | F1@K | PRC | F1@K |
| 1 | KD-ResNet | 0.792 | 0.793 | 0.546 | 0.559 | 0.581 | 0.444 | 0.538 | 0.588 | 0.161 | 0.187 |
| | KDAlign-ResNet | **0.805** | **0.800** | **0.642** | **0.581** | **0.696** | **0.575** | **0.601** | **0.643** | **0.201** | **0.266** |
| 3 | KD-ResNet | 0.643 | 0.651 | **0.629** | 0.612 | 0.560 | 0.422 | 0.595 | 0.565 | 0.156 | 0.188 |
| | KDAlign-ResNet | **0.744** | **0.742** | 0.613 | **0.645** | **0.676** | **0.592** | **0.621** | **0.654** | **0.207** | **0.261** |
| 5 | KD-ResNet | 0.812 | 0.780 | 0.682 | **0.667** | 0.517 | 0.410 | 0.600 | 0.645 | 0.168 | 0.194 |
| | KDAlign-ResNet | **0.834** | **0.806** | **0.691** | 0.645 | **0.699** | **0.577** | **0.668** | **0.694** | **0.207** | **0.263** |
| 10 | KD-ResNet | 0.715 | 0.767 | 0.633 | 0.612 | 0.485 | 0.506 | 0.706 | 0.700 | 0.171 | 0.191 |
| | KDAlign-ResNet | **0.848** | **0.768** | **0.659** | **0.656** | **0.604** | **0.594** | **0.770** | **0.734** | **0.209** | **0.262** |
| **Average** | KD-ResNet | 0.741 | 0.748 | 0.623 | 0.613 | 0.536 | 0.446 | 0.610 | 0.625 | 0.164 | 0.190 |
| | KDAlign-ResNet | **0.808** | **0.779** | **0.651** | **0.632** | **0.669** | **0.585** | **0.665** | **0.681** | **0.206** | **0.263** |

- We observe that our proposed KDAlign framework consistently enhances the performance of representative WSAD methods. For example, the KDAlign-PReNet method exhibits a 104.53% improvement over PReNet on the Satellite dataset, climbing from the last rank (excluding typical AD methods) to the second rank. In addition, DeepSAD, REPEN and DevNet, which introduced KDAlign, also have nearly doubled improvements on the Amazon and Cardiotocography data sets. Even when FeaWAD and ResNet already demonstrate commendable results, the KDAlign framework still manages to further boost their performance. In some isolated cases, KDAlign fails to enhance the performance of baseline methods. The reason may be that the unseen anomalies learned by the original baseline methods overlap with the anomalies covered by knowledge, and aligning them might distort the original data representation.

- We find that the best-performing methods on each dataset are based on the KDAlign framework. This indicates that KDAlign can not only improve the performance of baseline methods but also holds the potential to provide new state-of-the-art results in weakly supervised settings.

In addition, we also use Table 3 to present the experimental results of KDAlign and representative baselines with respect to AUCPR when the number of labeled anomalies is 1, 3, or 5. The results for F1@K will be shown in the appendix.

### 4.3 Ablation Study (RQ2)

We present the results of our Ablation Study in Table 4, with respect to AUPRC and F1@K. Concretely, we compare the model performance of KDAlign-ResNet and KD-ResNet across five datasets under four WSAD settings, where the KD-ResNet introduces knowledge without the knowledge-data alignment and the label anomalies of each dataset are respectively 1,3,5 and 10. According to the experimental results, we can clearly find that KDAlign-ResNet outperforms KD-ResNet in almost all settings. Besides, we observe that the standard deviation of KDAlign-ResNet across the four settings is significantly lower than that of KD-ResNet. This suggests that the performance of KDAlign-ResNet remains relatively consistent as the number of labeled samples varies from 1 to 10, reflecting the robustness of the KDAlign framework.

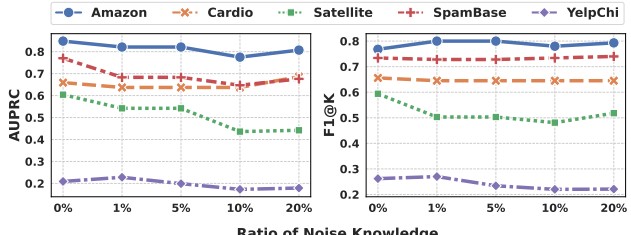

**Figure 3: Noisy knowledge study on KDAlign-ResNet.**

### 4.4 Impact of Noisy Knowledge (RQ3)

We use Figure 3 to illustrate the impact of noisy knowledge on the performance of KDAlign-ResNet. By 'noisy knowledge', we refer to the incompletely correct rules that will incorrectly judge some samples, leading to detection errors. Specifically, we investigate the performance of KDAlign-ResNet across five datasets under four noisy settings. The ratio of noise knowledge pertains to the ratio of incompletely correct rules to the total rules. From our experiments, we make the following observations: Compared to the setting without noise, we find that as the ratio increases, the performance of KDAlign-ResNet does not fluctuate significantly, except for the Satellite dataset. The reason might be the knowledge with noise happens to cover some unseen important anomaly scenarios, which in turn results in a decline in model performance. It is worth noting that compared with Table 2, even when impacted by noisy knowledge, the performance of KDAlign-ResNet remains superior to that of ResNet.

## 5 RELATED WORK

### 5.1 Weakly Supervised Anomaly Detection

Weakly supervised anomaly detection aims to train an effective AD model with limited labeled anomaly samples and extensive unlabeled data. Early studies [40, 48, 49] on WSAD primarily involved designing a feature extractor based on unsupervised AD algorithms and then learning a supervised classifier using labeled data such as eepSAD [49] and REPEN [40]. Recent studies [36, 42, 43, 65] focus on designing end-to-end deep framework. For example, DevNet [43] utilizes a prior probability and a margin hyperparameter to enforce obvious deviations in anomaly scores between normal and abnormal data. FeaWAD [65] incorporates the DAGMM [66] network

architecture with the deviation loss. PReNet [42] formulates the scoring function as a pairwise relation learning task.

Another research line utilizes active learning or reinforcement learning to reduce the cost of obtaining anomaly labels. For instance, AAD [15] leverages the active learning technique, which operates in an interactive loop for data exploration and maximizes the total number of true anomalies presented to the expert under a query budget. DPLAN [44] considers simultaneously exploring both limited labeled anomaly examples and scarce unlabeled anomalies to extend the learned abnormality, leading to the joint optimization of both objectives.

In contrast to above studies, our work introduces rule knowledge to supplement the limited anomaly samples. Similar to label annotations, such knowledge also contains human supervision, but has been largely overlooked.

### 5.2 Neural-symbolic Systems

The symbolic system excels in leveraging knowledge, while the neural system is adept at harnessing data. Both knowledge and data play a pivotal role in decision-making processes. There is a burgeoning interest among AI researchers to fuse the symbolic and neural paradigms, aiming to harness the strengths of both [18, 19, 28, 51, 57]. When juxtaposing neural-symbolic systems against purely neural or symbolic ones, three aspects come to the fore [60]. First is the Efficiency. Neural-symbolic models can expedite computations, making them suitable for reasoning on vast data sets. Second is the Generalization. These systems are not solely reliant on extensive labeled datasets, endowing them with impressive generalization capabilities. By integrating expert or background knowledge, neural-symbolic models can compensate for sparse training data, achieving commendable performance without sacrificing generalizability. Third is the interpretability. Neural-symbolic architectures offer transparency in their reasoning, enhancing their interpretability [60]. Such transparency is invaluable in fields like medical image analysis, where stakeholders require both the outcome and an understanding of the decision-making rationale [60]. In general, the neural-symbolic system is a promising approach to effectively simultaneously leverage knowledge and data for decision-making processes. However, its potential in weakly supervised anomaly detection has yet to be explored.

## 6 CONCLUSION AND FUTURE WORK

In this paper, we study the problem of weakly supervised anomaly detection and propose a novel WSAD framework named KDAlign, which reformulates knowledge incorporation as knowledge-data alignment, adopts OT for effectively resolving knowledge-data alignment, and finally supplements the limited anomaly samples to improve the performance of WSAD models. We extensively conduct experiments on five real-world datasets and the experimental results demonstrate that our framework outperforms the other competitors.

For the future, we plan to extend our work in following directions: (1) Extend to graph domain [27]; (2) Introduce other OT methods, such as [54]; (3) Improve explainability.

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

## A  KNOWLEDGE ACQUISITION

Rule knowledge usually widely exists in industry [64]. However, hindered by concerns for industrial safety and privacy, procuring traditional rule knowledge from the industry poses challenges. Therefore, it is necessary to find an alternate way to simulate the industrial rule knowledge. We find the decision tree method is a promising way [4, 46], where every decision path can be regarded as a rule knowledge. First, the representation format of decision paths is the same as industrial rule knowledge, often manifesting as if/else statements. Second, decision paths are conveniently accessible—for instance, we can extract decision paths from well-trained decision trees.

Specifically, referring to Fig. 4, we assign three steps to acquire rule knowledge based on decision tree models: **Step 1:** Given a collection of $m$ samples $\mathbf{X} = \{x_1, ..., x_m\} \in \mathbb{R}^{m \times d}$ and the binary ground truth labels $\mathbf{y} = \{y_1, ..., y_m\} \in \{0, 1\}^m$, we train $r$ decision trees; **Step 2:** For the $r$ trained decision trees, we extract *all-right anomaly paths* (knowledge set) $\mathbf{R} = \{r_1, ..., r_s\}$ included in them as the rule knowledge. An *all-right anomaly path* means that the labels of specific samples in $\mathbf{X}$ passing the decision path are all anomalous.

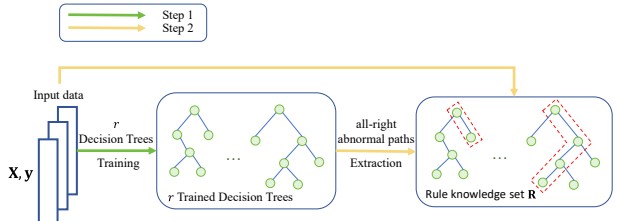

**Figure 4: Knowledge Acquisition: Using decision trees to simulate industrial rule knowledge.**

## B  KNOWLEDGE ENCODER

**Preliminaries:** This is a brief introduction to d-DNNF, which is used in Knowledge Encoder. A formula that is a conjunction of clauses (a disjunction of literals) is in the Conjunctive Normal Form (CNF). Let $S$ be the set of propositional variables. A sentence in Negation Normal Form (NNF) is defined as a rooted directed acyclic graph (DAG) where each leaf node is labeled with True, False, $s,$ $or \neg s, s \in S$; and each internal node is labeled with $\wedge$ or $\vee$ and can have discretionarily many children. Deterministic Decomposable Negation Normal Form (d-DNNF) [12, 14] further imposes that the representation is: (i) **Deterministic**: An NNF is deterministic if the operands of $\vee$ in all well-formed boolean formula in NNF are mutually inconsistent; (ii) **Decomposable**: An NNF is decomposable if the operands of $\wedge$ in all well-formed boolean formula in the NNF are expressed on a mutually disjoint set of variables. Opposite to CNF and more general forms, d-DNNF has many satisfactory tractability properties (e.g., polytime satisfiability and polytime model counting). Because of having tractability properties, it is appealing for complex AI applications to adopt d-DNNF [11].

In the paper, we mentioned the use of a **Knowledge Encoder** module to map propositional formulae into an embedding space. Concretely, we utilize the d-DNNF graph structure to represent

a propositional formula $f_i$ and then apply a multi-layer Graph Convolutional Network [31] as an encoder to project the formula, $f_i$. In the following paragraphs, we further detail the **Knowledge Encoder** module. Note that the **Knowledge Encoder** $\phi_F(\cdot)$ is trained before KDAlign framework.

The input for training $\phi_F(\cdot)$ consists of specialized d-DNNF graphs which contribute to enhanced symbolic (knowledge) embeddings. These graphs are built from formulae that have been restructured based on decision paths. To construct the specific graphs based on these formulae, we first change the formulae in CNF and then use **c2d** to compile these formulae in d-DNNF [12–14]. For example, based on Formula '$p_1 \wedge p_2 \Rightarrow q$' in Section 3.2, we construct a CNF expression by Formula. (7). Then, after executing **c2d**, Formula. (7) can be expressed in d-DNNF shown by Formula. (8).

$$\neg p_1 \vee \neg p_2 \vee q \tag{7}$$

$$(\neg p_1 \wedge p_2) \vee \neg p_2 \vee q \tag{8}$$

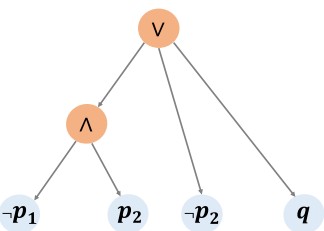

**Figure 5: The d-DNNF graph structure generated based on Formula.** (8).

Then a propositional formula can be represented as a directed or undirected graph $G = (V, E)$, consisting of $N$ nodes denoted by $v_i \in V$ and edges represented as $(v_i; v_j) \in E$. Individual nodes are either propositions (leaf nodes) or logical operators ($\wedge; \vee; \Longrightarrow$), where propositions are connected to their respective operators. Fig. 5 can help understand the concrete structure. In addition to the mentioned nodes, every graph, like Fig. 5, is further augmented by a global node linked to all other nodes. In $\phi_F(\cdot)$, the graphs are regarded as undirected graphs.

The layer-wise propagation rule of GCN is,

$$Z^{(l+1)} = \sigma(\tilde{D}^{-\frac{1}{2}} \tilde{A} \tilde{D}^{-\frac{1}{2}} Z^{(l)} W^{(l)}) \tag{9}$$

where $Z^{(l+1)}$ represent the learnt latent node embeddings at $l^{th}$ (note that $Z^{(0)} = X$), $\tilde{A} = A + I_N$ represents the adjacency matrix of the undirected graph $G$ with added self-connections through the identity matrix $I_N$. $\tilde{D}$ is a diagonal degree matrix with $\tilde{D}_{ii} = \sum_j \tilde{A}_{ij}$. The weight matrices for layer-specific training are $W^{(l)}$, and $\sigma(\cdot)$ represents the activation function. To more effectively capture the semantics conveyed through the graphs, the $\phi_F(\cdot)$ function incorporates two additional adjustments: heterogeneous node embeddings and semantic regularization, as cited in [58]. The concrete code implementation is accessible at https://github.com/ZiweiXU/LENSR.

**Table 5: Performance comparison between representative baselines and KDAlign under the setting of 1, 3, or 5 labeled anomalies w.r.t F1@K. '-' indicates that the PReNet model can not handle setting of only 1 labeled anomaly samples.**

| Model | Amazon | | | Cardio | | | Satellite | | | SpamBase | | | YelpChi | | |
|---|---|---|---|---|---|---|---|---|---|---|---|---|---|---|---|
| | 1 | 3 | 5 | 1 | 3 | 5 | 1 | 3 | 5 | 1 | 3 | 5 | 1 | 3 | 5 |
| KNN | 0.052 | 0.045 | 0.071 | 0.204 | 0.215 | 0.226 | 0.357 | 0.357 | 0.357 | 0.420 | 0.417 | 0.411 | 0.150 | 0.149 | 0.149 |
| SVM | 0.045 | 0.026 | 0.026 | 0.269 | 0.280 | 0.215 | 0.308 | 0.279 | 0.318 | 0.431 | 0.386 | 0.440 | 0.162 | 0.153 | 0.171 |
| DT | 0.052 | 0.052 | 0.097 | 0.194 | 0.194 | 0.226 | 0.355 | 0.357 | 0.355 | 0.426 | 0.426 | 0.420 | 0.150 | 0.150 | 0.150 |
| DevNet | **0.090** | 0.148 | 0.123 | 0.247 | 0.269 | 0.258 | **0.555** | **0.557** | 0.562 | 0.420 | 0.423 | 0.423 | **0.181** | 0.185 | 0.184 |
| KDAlign-DevNet | 0.071 | **0.747** | **0.581** | **0.528** | **0.436** | **0.502** | 0.449 | **0.641** | **0.686** | **0.489** | **0.461** | **0.494** | 0.156 | **0.200** | **0.195** |
| PReNet | - | 0.277 | 0.568 | - | 0.419 | 0.581 | - | 0.386 | 0.244 | - | 0.669 | 0.754 | - | 0.173 | 0.194 |
| KDAlign-PReNet | - | **0.658** | **0.600** | - | **0.634** | **0.591** | - | **0.597** | 0.641 | - | **0.703** | 0.749 | - | **0.221** | **0.224** |
| DeepSAD | 0.110 | 0.077 | 0.161 | 0.387 | 0.258 | 0.247 | 0.582 | **0.577** | 0.548 | 0.651 | 0.629 | 0.594 | 0.149 | 0.221 | 0.221 |
| KDAlign-DeepSAD | **0.355** | **0.484** | **0.368** | **0.624** | **0.495** | **0.581** | **0.641** | 0.495 | **0.643** | **0.706** | **0.723** | **0.689** | **0.156** | **0.248** | **0.259** |
| REPEN | 0.039 | 0.039 | 0.039 | 0.473 | 0.473 | 0.473 | 0.648 | 0.655 | 0.655 | 0.589 | 0.589 | 0.589 | **0.283** | **0.282** | **0.283** |
| KDAlign-REPEN | **0.289** | **0.289** | **0.289** | **0.631** | **0.631** | **0.631** | **0.720** | **0.720** | **0.720** | **0.608** | **0.608** | **0.608** | 0.181 | 0.180 | 0.209 |
| FeaWAD | 0.677 | 0.368 | 0.710 | 0.570 | **0.548** | **0.656** | 0.472 | 0.457 | **0.538** | **0.677** | 0.703 | 0.726 | 0.199 | 0.204 | 0.235 |
| KDAlign-FeaWAD | **0.729** | **0.677** | **0.774** | **0.602** | 0.505 | 0.581 | **0.623** | **0.650** | 0.501 | 0.640 | **0.754** | **0.740** | **0.245** | **0.228** | **0.266** |
| ResNet | 0.781 | 0.658 | 0.774 | **0.602** | 0.645 | **0.677** | 0.340 | 0.369 | 0.369 | 0.620 | 0.586 | 0.603 | 0.195 | 0.190 | 0.207 |
| KDAlign-ResNet | **0.800** | **0.742** | **0.806** | 0.581 | 0.645 | 0.645 | **0.575** | **0.592** | **0.577** | **0.643** | **0.654** | **0.694** | **0.266** | **0.261** | **0.263** |

**Table 6: Optimal Parameter of KDAlign-FeaWAD**

| Dataset Name | Epoch | Layers | Learning Rate | Hidden Dimension | Rule Weight | Activation |
|---|---|---|---|---|---|---|
| Amazon | 20 | 2 | 0.001 | 32 | 0.01 | ReLU |
| Cardiotocography | 20 | 2 | 0.01 | 32 | 0.01 | ReLU |
| Satellite | 20 | 3 | 0.001 | 64 | 0.05 | ReLU |
| SpamBase | 20 | 3 | 0.001 | 64 | 0.05 | ReLU |
| YelpChi | 100 | 2 | 0.01 | 32 | 0.05 | ReLU |

**Table 7: Optimal Parameter of KDAlign-ResNet**

| Dataset Name | Epoch | Learning Rate | Blocks | Hidden Dimension | Rule Deight | Main Dimension | Dropout First | Dropout Second |
|---|---|---|---|---|---|---|---|---|
| Amazon | 50 | 0.01 | 3 | 256 | 0.1 | 192 | 0.2 | 0 |
| Cardiotocography | 50 | 0.01 | 3 | 128 | 0.01 | 64 | 0.2 | 0 |
| Satellite | 200 | 0.01 | 3 | 128 | 3 | 128 | 0.2 | 0 |
| SpamBase | 50 | 0.01 | 2 | 128 | 0.01 | 64 | 0.2 | 0 |
| YelpChi | 50 | 0.01 | 3 | 256 | 3 | 64 | 0.2 | 0 |

## C  IMPLEMENTATION DETAILS

**Hardware Specifications**. All our experiments were carried out on a Linux server equipped with AMD EPYC 7763 64-Core Processor, 503GB RAM, and eight NVIDIA RTX4090 GPUs with a total of 192G memory.

**Hyperparameter Settings**. Table 6 and Table 7 respectively show our optimal hyperparameter settings of KDAlign-FeaWAD and KDAlign-ResNet utilized in our experiments clearly, which are trained on 10 labeled anomaly samples.

