# OpenReview forum: "Weakly Supervised Anomaly Detection via Knowledge-Data Alignment"
_ACM.org/TheWebConf/2024/Conference — TheWebConf24 Oral_

### Official Review · Reviewer_AMY2 · 2023-11-24

**Novelty:** 5
**Technical Quality:** 6

**Review:**

This work proposes a novel framework for weakly supervised anomaly detection via **knowledge-data alignment**. The authors claim that their framework can incorporate rule knowledge, derived from human experts, to supplement the limited labeled anomaly samples and improve the performance of deep learning models. The authors use **optimal transport** techniques to align knowledge and data in a high-dimensional embedding space and introduce an additional loss term to the original objective function of weakly supervised anomaly detection methods. The authors conduct experiments on **five** real-world datasets and demonstrate that their framework outperforms several baselines and achieves state-of-the-art results.

**Questions:**

1. Could you provide more details about the **knowledge encoder** used in the KDAlign framework inside the main text? How were they implemented and trained?

2. Could you provide some **qualitative examples** or **visualizations** of the knowledge-data alignment? What does the alignment look like in the high-dimensional embedding space, and how does it change during the training process?

3. Could you discuss the **limitations** of the KDAlign framework and the potential **future work**? For example, how to handle noisy or incomplete rule knowledge, how to scale up to large datasets or complex models, or how to incorporate other types of weak supervision such as partial labels or pairwise constraints?

**Reviewer Confidence:**

1: The reviewer's evaluation is an educated guess

**Scope:**

3: The work is somewhat relevant to the Web and to the track, and is of narrow interest to a sub-community

---

### Official Review · Reviewer_AQ1S · 2023-11-24

**Novelty:** 5
**Technical Quality:** 6

**Review:**

This paper proposes KDAlign to integrate rule knowledge, typically summarized by human experts, to supplement limited labeled data in a weakly supervised anomaly detection problem.

Strengths

1. This paper is easy to understand.
2. The proposed idea of utilizing knowledge for anomaly detection makes sense and is expected to work well in other scenarios.
3. Extensive experiments were conducted to demonstrate the effectiveness of the proposed approach.

Weaknesses

1. There are unclear details on the proposed approach. Some parts of the approach, such as building a knowledge graph, should be better explained in the main paper, not only in the appendix. Please refer to the questions below.
2. The main experimental setting is too favorable to the proposed approach, as the authors delete anomaly samples that match rules from the training set. It is uncertain if the proposed approach can work in more challenging settings.

**Questions:**

Questions

1. Why do we need to "align" the data and knowledge in the first place? Is alignment the only way to incorporate knowledge into predictions?
2. If the sizes of the data and knowledge sets are different, how can we define the knowledge-data alignment M? What if multiple rules can be applied to a single data point?
3. What is $E_T$? I can't find its definition in the text.
4. How is weakly-supervised anomaly detection (AD) different from semi-supervised AD?
5. Is the proposed approach the first attempt to use such knowledge in anomaly detection?

**Reviewer Confidence:**

3: The reviewer is confident but not certain that the evaluation is correct

**Scope:**

3: The work is somewhat relevant to the Web and to the track, and is of narrow interest to a sub-community

---

### Official Review · Reviewer_NGBz · 2023-11-27

**Novelty:** 3
**Technical Quality:** 4

**Review:**

This paper focuses on Weakly Supervised Anomaly Detection (WSAD) problem with limited labeled data. Specifically, the authors propose Knowledge-Data Alignment (KDAlign) to integrate rule knowledge for anomaly detection via transforming knowledge and then aligning with data in representation space. As a result, the alignment loss term is proposed using optimal transport as an additional loss. Experiments demonstrate the effectiveness of KDAlign for real world datasets and noisy knowledge setting.

Strong points
1.	This paper is well-written and easy to follow.
2.	The research problem to integrate knowledge rule is well-motivated and the knowledge-data alignment solution is simple and natural.

Weak points
1.	The technical novelty is limited. Even though the Knowledge-Data Alignment idea is interesting, the knowledge rules, as an auxiliary information, have been integrated in many papers. The proposed method is not new from my pool. Additionally, If we have knowledge rule, can we use these rules to generate pseudo labels and the train the model? Is there any specific reason to encode knowledge instead of generating labels? What’s the performance comparison results?
2.	The rationale of superiority of KDAlign for noisy knowledge is still unclear to me. Why can KDAlign tackle noisy knowledge during training? I understand it can tackle noisy knowledge in inference for the well-trained model, but what if noisy knowledge during training? Can you elaborate more on the points/reason tackling noisy knowledge in training? Additionally, there is not clear relation between performance improvement and noisy ratio. Is the performance improvement larger for high noise? What’s the maximum noise ratio that KDAlign can tackle?
3.	Why optimal transport? In Lines 439-440, the authors attribute the robustness over noise knowledge into OT due to the global perspective. What is rationale behind that? From my understanding, there are multiple distribution distance metrics to measure such global distance, such as mutual information, MSE, KL divergence etc. Any insights behind OT? If yes, more ablation study on distance metrics should be conducted.

**Questions:**

1.	The technical novelty is limited. Even though the Knowledge-Data Alignment idea is interesting, the knowledge rules, as an auxiliary information, have been integrated in many papers. The proposed method is not new from my pool. Additionally, If we have knowledge rule, can we use these rules to generate pseudo labels and the train the model? Is there any specific reason to encode knowledge instead of generating labels? What’s the performance comparison results?
2.	The rationale of superiority of KDAlign for noisy knowledge is still unclear to me. Why can KDAlign tackle noisy knowledge during training? I understand it can tackle noisy knowledge in inference for the well-trained model, but what if noisy knowledge during training? Can you elaborate more on the points/reason tackling noisy knowledge in training? Additionally, there is not clear relation between performance improvement and noisy ratio. Is the performance improvement larger for high noise? What’s the maximum noise ratio that KDAlign can tackle?
3.	Why optimal transport? In Lines 439-440, the authors attribute the robustness over noise knowledge into OT due to the global perspective. What is rationale behind that? From my understanding, there are multiple distribution distance metrics to measure such global distance, such as mutual information, MSE, KL divergence etc. Any insights behind OT? If yes, more ablation study on distance metrics should be conducted.

**Reviewer Confidence:**

3: The reviewer is confident but not certain that the evaluation is correct

**Scope:**

3: The work is somewhat relevant to the Web and to the track, and is of narrow interest to a sub-community

---

### Official Review · Reviewer_s7w9 · 2023-11-29

**Novelty:** 5
**Technical Quality:** 4

**Review:**

This paper studies the problem of weakly supervised anomaly detection by integrating rule knowledge via knowledge-data alignment. In general the studied problem is interesting and the proposed solution sounds reasonable. My major concern is the clarity of methodology, especially the calculation of optimal transport. Detailed comments are listed as follows.

Strengths
1. The proposed intuitive of using rule knowledge to aid weakly supervised anomaly detection is well justified;
2. The proposed solution provides some technical contribution to solve this problem;
3. The empirical results supports the claims.

Weaknesses
1. The clarify of methodology section needs to be improved. I am not particularly familiar with this topic, but as a general reader, I cannot justify the validity of several key techniques given their current forms. For instance, in line 354-355, it is unclear about how the propositional formula in F is transformed into graphs and the authors should provide more illustration to this step. Section 3.3 is also presented in an abstract manner without relating to X and F, which is hard to follow: what is the form of $S(f_i, x_j)$? How is $C$ calculated? What are $u$ and $v$ in the context of $E_X$ and $E_F$?
2. Some claims are not well justified. In the discussion of noisy rule, in line 148-150, the authors mentioned "when a sample matches a noisy rule, the distance of that sample to some other closely related rules will be farther". Is the claim supported by any evidence?

**Questions:**

1. Are the rules generated on training, validation or test data? What does line 573 "delete the anomaly samples that match rules from the training set" mean?
2. What does it mean by KD-ResNet introduces knowledge without the knowledge-data alignment? Then how is the knowledge introduced?
3. How are baselines influenced by noisy knowledge?
4. How sensitive is the method to the hyperparameter $\lambda$?

**Reviewer Confidence:**

2: The reviewer is willing to defend the evaluation, but it is likely that the reviewer did not understand parts of the paper

**Scope:**

3: The work is somewhat relevant to the Web and to the track, and is of narrow interest to a sub-community

---

### Decision · Program_Chairs · 2024-01-22

**Decision:**

Accept (Oral)

**Comment:**

This paper proposes an approach for weakly supervised anomaly detection using knowledge-data alignment. The proposal allows incorporating rule knowledge from human experts to supplement limited labeled anomalies.

 Overall, reviewers rated this paper generally favorably in both technical quality and novelty (especially AQ1S and AMY2). The main technical concerns raised pre-rebuttal were largely addressed by authors in detailed responses. In particular, reviewers raised several evaluation gaps including comparisons with pseudo-labeling, noise ratios and hyperparameter sensitivity, which were satisfactorily resolved given discussion.

 I encourage authors to be mindful in adapting the requested updates to the final version.